

# Morphology of inner cell mass: a better predictive biomarker of blastocyst viability

Sargunadevi Sivanantham[1], Mahalakshmi Saravanan[2], Nidhi Sharma[3], Jayashree Shrinivasan[3] and Ramesh Raja[4]

[1] Department of IVF, ARC International Fertility and Research Centre, Chennai, Tamil Nadu, India
[2] Department of Reproductive Medicine, ARC International Fertility and Research Centre, Chennai, Tamil Nadu, India
[3] Department of Obstetrics and Gynaecology, Saveetha Medical College, Chennai, Tamil Nadu, India
[4] Department of Andrology and Reproductive Medicine, Chettinad Hospital and Research Institute, Chennai, Tamil Nadu, India

## ABSTRACT

**Background**. Transfer of embryos at the blastocyst stage is one of the best approaches for achieving a higher success rate in *In vitro* fertilization (IVF) treatment as it demonstrates an improved uterine and embryonic synchrony at implantation. Despite novel biochemical and genetic markers proposed for the prediction of embryo viability in recent years, the conventional morphological grading of blastocysts remains the classical way of selection in routine practice. This study aims to investigate the association between the morphological features of blastocysts and pregnancy outcomes.
**Methods**. This prospective study included women undergoing single or double frozen blastocyst transfers following their autologous cycles in a period between October 2020 and September 2021. The morphological grades (A—good, B—average, and C—poor) of inner cell mass (ICM) and trophectoderm (TE) of blastocysts with known implantation were compared to assess their predictive potential of pregnancy outcome. It was further explored by measuring the relationship between the two variables using logistic regression and receiver operating characteristic (ROC) analysis.
**Results**. A total of 1,972 women underwent frozen embryo transfer (FET) cycles with a total of 3,786 blastocysts. Known implantation data (KID) from 2,060 blastocysts of 1,153 patients were subjected to statistical analysis, the rest were excluded. Implantation rates (IR) from transfer of ICM/TE grades AA, AB, BA, BB were observed as 48.5%, 39.4%, 23.4% and 25% respectively. There was a significantly higher IR observed in blastocysts with ICM grade A ($p < 0.001$) than those with B irrespective of their TE scores. The analysis of the interaction between the two characteristics confirmed the superiority of ICM over TE as a predictor of the outcome. The rank biserial correlation value for ICM was also greater compared to that of TE (0.11 vs 0.05).
**Conclusion**. This study confirms that the morphology of ICM of the blastocyst is a stronger predictor of implantation and clinical pregnancy than that of TE and can be utilized as a biomarker of viability.

Corresponding author
Sargunadevi Sivanantham,
deviivf@yahoo.co.in

## INTRODUCTION

In assisted reproduction, the prediction of a successful outcome is dependent on the endometrium and the selection and replacement of embryos with higher implantation potential. The development of sophisticated culture media systems offers effective extended culture and utility of embryos up to the blastocyst stage. The first ever reported IVF pregnancy was from a blastocyst transfer (*Hard arson, Van Landuyt & Jones, 2012*). It has been shown that better implantation, pregnancy, and live birth rates can be achieved from blastocyst transfers (*Gardner & Lane, 1997*; *Blake et al., 2007*; *Papanikolaou et al., 2008*). A greater embryo endometrial synchrony is demonstrated when embryos are transferred at the blastocyst stage (*Ahlström et al., 2011*; *Chen et al., 2014*; *Subira et al., 2016*). The true viability of the embryo can only be assessed post embryonic genome activation (*Gardner & Balaban, 2016*).

The defining aspects of blastocyst—the formation of blastocoel cavity and cell differentiation between ICM and TE are considered the points of morphological assessment providing a composite score (*Hard arson, Van Landuyt & Jones, 2012*). Conventional grading of blastocysts comprises of three notable parameters (i) expansion of the blastocoelic cavity (ii) the characteristics of the ICM and (iii) that of TE. In routine practice, *in vitro* development of preimplantation embryos is assessed by microscopic examination at specific time intervals. Different grading systems for blastocyst have been developed (*Puga-Torres, Blum-Rojas & Blum-Narváez, 2017*).

The morphological scores of blastocyst characteristics are closely associated with the success rates. ICM has been suggested as a better predictor of implantation (*Alpha Scientists in Reproductive Medicine and ESHRE Special Interest Group of Embryology, 2011*; *Irani et al., 2017*; *Licciardi et al., 2015*). However other groups have also reported the predictive strength of TE on the implantation, miscarriage, and delivery rates (*Hill et al., 2013*; *Honnma et al., 2012*). The superiority of either of these morphological features in correlation with clinical pregnancy is still under debate.

In recent decades, with the advent of modern technologies, morphokinetics, ploidy, and physiology of human preimplantation embryos have been studied extensively. These technologies brought in ways to identify novel viability markers within a cohort of embryos for transfer or to cryopreserve for subsequent cycles. The methods could be invasive or non-invasive by nature. The usefulness and limitations of some of these methods are discussed below.

Time lapse technology (TL) has emerged as one of the most useful tools to study the developmental patterns of embryos using image capture technology utilizing uninterrupted culture conditions. These non-invasive selection models may not be directly transferrable between clinics or patient groups (*Campbell et al., 2013*). The information on TL markers from various retrospective studies is heterogeneous and ambiguous (*Kirkegaard, Agerholm & Ingerslev, 2012*; *Kirkegaard et al., 2016*). It is still not clear whether the superiority is based on uninterrupted culture condition alone or its combination with selection algorithms (*Fréour et al., 2015*; *Liu et al., 2016*). Moreover, TL adds additional costs to patients and clinics. A recently published prospective randomized control trial (RCT) on

TL *vs* morphology study found no significant improvement in the clinical pregnancy rate (CPR) from single embryo transfers between the groups (*Ahlström et al., 2022*). There is a need for high-quality data from well-controlled large randomized trials to confirm its benefits in the area of reproductive medicine (*Bhide et al., 2020*; *Armstrong et al., 2019*; *Armstrong et al., 2015*; *Fishel et al., 2018*; *Sayed et al., 2020*).

Automation in TL using artificial intelligence (AI), either by computer vision technology or by deep learning frameworks serves as an embryo ranking tool. Though it helps considerably in laboratory decisions of choice of embryos, their interpretability and selection bias within the models remain the main concerns. There is a need for prospective trials and long-term follow-up of babies born before their clinical application (*Kragh & Karstoft, 2021*; *Mihdi Afnan et al., 2021*).

Preimplantation genetic testing (PGT) encompasses methods of testing the embryos for severe inherited conditions or chromosomal abnormality by removal of one or more cells from the developing embryo (*Leaver & Wells, 2019*). In PGT for aneuploidies (PGT A), a few cells of TE are biopsied for testing, and blastocysts with no chromosomal abnormality are replaced to achieve healthy live births. It has been initially reported that PGT improved the live birth rate in patients with advanced age, implantation failures, and loss. Currently, the available pieces of evidence do not support the routine use of PGT A as a selection model. The reliability of TE sampling to reflect the genetic materials of ICM is under debate (*Esfandiari, Bunnell & Casper, 2016*; *Orvieto et al., 2016*; *Gleicher & Orvieto, 2017*; *Gleicher et al., 2017*; *Chuang et al., 2018*). PGT A adds more cost to IVF treatment and needs further reconfirmation with prenatal diagnostic procedures (*Ly, Agarwal & Nagy, 2011*). Embryo biopsy requires experienced lab personnel and the damage to the embryo at biopsy must also be considered.

A recent large RCT showed no benefit of PGT A over the standard morphological assessment of embryos for women above 35 years of age (*Kang et al., 2016*). Furthermore, more than 100 live births have been reported from the transfer of mosaic embryos and it raises concerns about the misinterpretation of results (*Abhari & Kawwass, 2021*). Thus, large prospective randomized control trials are necessary to improve the practice of invasive PGT A as a viable embryo selection strategy (*Kane, 2016*).

Attention has been given in recent years to the development of non-invasive PGT (niPGT) methods such as analysis of extra-embryonic DNA from aspiration of blastocoel fluid or spent culture medium. Ongoing studies are attempting to verify the source of DNA is reliable and represents the genetic status of the embryo. Carefully designed clinical investigations and optimization of analysis methods are needed to confirm the accuracy of niPGT in clinical use (*Leaver & Wells, 2019*; *Ben-Nagi et al., 2019*).

Omics technologies are methods that incorporate a series of complex processes that analyse several molecules in a single biological sample using advanced biotechnology techniques. The analysis of embryonic and endometrial components required during implantation can be well studied by integrative omics models (*Koroknai et al., 2019*). Quantification of embryonic viability by metabolic and protein profiling of the spent culture medium can optimize the embryo selection and the IVF outcome. However, the ability of embryos to adapt to different culture conditions and oxygen concentrations, difficulty in

interpretation of data between centres using nonstandardized culture conditions, cost and complexity of the equipment, and the turnaround time need to be considered before its application into a clinical setting (*Omics in Reproductive Medicine, 2013*).

Although numerous morphological and biochemical markers have been suggested, evaluation and selection by morphological assessment remain the standard method of prioritizing blastocysts for transfer (*Zhao, Yu & Zhang, 2018*).

This study aims to demonstrate the benefit of conventional morphological grading for viable embryo strategy, the contribution of ICM and TE, and their interdependency in serving as biomarkers of implantation and clinical pregnancy in frozen blastocyst cycles.

## MATERIALS & METHODS

This multicentre prospective study included women undergoing FET cycles either as a single embryo transfer (SET) or a double embryo transfer (DET) from October 2020 to September 2021 and was approved by the Institutional Ethics Committee at Saveetha Medical College Hospital, Chennai, India (No. 006/09/2020/IEC/SMCH). The patient information sheet was filled in and written consent forms were obtained from all the participants of this study.

### Patient inclusion and exclusion criteria

Women between 25 and 37 years of age who underwent IVF treatment with Intracytoplasmic sperm injection (ICSI) as the choice of insemination, having metaphase II oocytes equal to or more than 6, sperm with less than 30% DNA fragmentation, blastocysts frozen, and transferred in the subsequent FET were included in the study.

Patients with body mass index >28, polycystic ovaries, hypo or hyperthyroidism, diabetes, endometriosis, AMH<1.5, men with sperm DNA fragmentation above 30%, cycles with insemination by conventional IVF method, and fresh embryo transfers were excluded from the study.

### Clinical and laboratory protocols
#### Ovarian stimulation

GnRH antagonist with gonadotropins was used for ovarian stimulation. On day 2 of the menstrual cycle, based on age, body mass index, and antral follicle count of patients, follicles were stimulated for 11–13 days using recombinant FSH—1200 IU with a starting dose of 225–300 IU and stepped down. GnRH antagonist Cetrorelix acetate—0.25 mg was administered on day 5 or 6 based on lead follicle size and serum oestradiol levels. Response to stimulation was monitored by transvaginal ultrasounds and gonadotropin dosage was adjusted accordingly.

Final maturation was triggered with recombinant human chorionic gonadotropin (HCG—250 mcg) or Triptorelin acetate—0.2 mg. Transvaginal oocyte retrieval was carried out after 35–36 h of the trigger.

### Laboratory culture and cryopreservation

Collected oocytes were cultured in single step culture medium (Onestep, Vitromed, Germany). Cumulus complexes were removed using hyaluronidase enzyme (Hyadase 80

IU/ml; Vitromed, Jena, Germany) within two hours from the time of collection. ICSI was performed within 40 h from the time of trigger. Fertilization and embryo development was assessed at specific time intervals. The standard protocol included assessment of fertilization at 17–18 h post-ICSI followed by cleavage checks every 24 h up to day 6 of culture. Cultures were maintained under conditions of 37 °C, 6% CO2, and 5% O2 until day 6 of the development single step medium (Onestep; Vitromed, Jena, Germany) with no refreshment of the medium on day 3. Suitable blastocysts were frozen on days 5 and 6 of development by the vitrification method (Kitazato, Tokyo, Japan).

## Blastocyst grading

The evaluation of blastocysts was based on Gardener and Schoolcraft's blastocyst grading scale (*Gardner et al., 2004*) which enumerates the degree of expansion and hatching represented by a numerical value (1–6) followed by the appearance of the key features, the ICM then the TE in letters (A–C). The blastocysts were graded by two competent embryologists in each clinic. Based on the number and quality of the embryos, suitable blastocysts were selected and vitrified.

The degree of blastocoel expansion was categorized as the following; (1) the early blastocyst where blastocoel fluid fills less than half of the cavity and the cell types are not distinguishable; (2) the blastocyst where blastocoel fluid fills more than half of the cavity with a visualization of ICM and TE; (3) the expanding blast with a large fully formed cavity with the start of thinning of zona pellucida; (4) the expanded blast where the blastocyst expands in size and the zona pellucida thins out; (5) the hatching blast with few cells from the blastocyst hatch out of zona pellucida; (6) the hatched blast with complete extrusion of the blastocyst from the zona pellucida (Fig. S1).

ICM of the blastocyst was classified as A where many cells form an intact mass within the blastocyst cavity, B where few cells were present or loosely attached, and C where very few cells were present or cells were absent.

Likewise, TE was graded A when many cells formed a cohesive epithelium, B when few cells were present without a cohesive epithelium, and C when sparse cells were present.

The possible combinations of ICM/TE morphology irrespective of degree of expansion at blastocyst grading were shown as follows; AA, AB, AC, BA, BB, BC, CA, CB, and CC (Fig. 1). Additional images of blastocysts of above mentioned grades were enclosed in Fig. S2.

## Endometrial preparation

The Hormone Replacement Treatment (HRT) protocol was used for endometrial preparation in frozen-thawed cycles. Oestradiol valerate was administered with an initial dose of 6 mg from day 2 of the menstrual cycle and tailored based on endometrial thickness. Once the desired thickness of 7–8 mm was achieved, oral progesterone 100 mg was started along with a vaginal progesterone suppository of 400 mg. FET was performed after five days of progesterone administration.

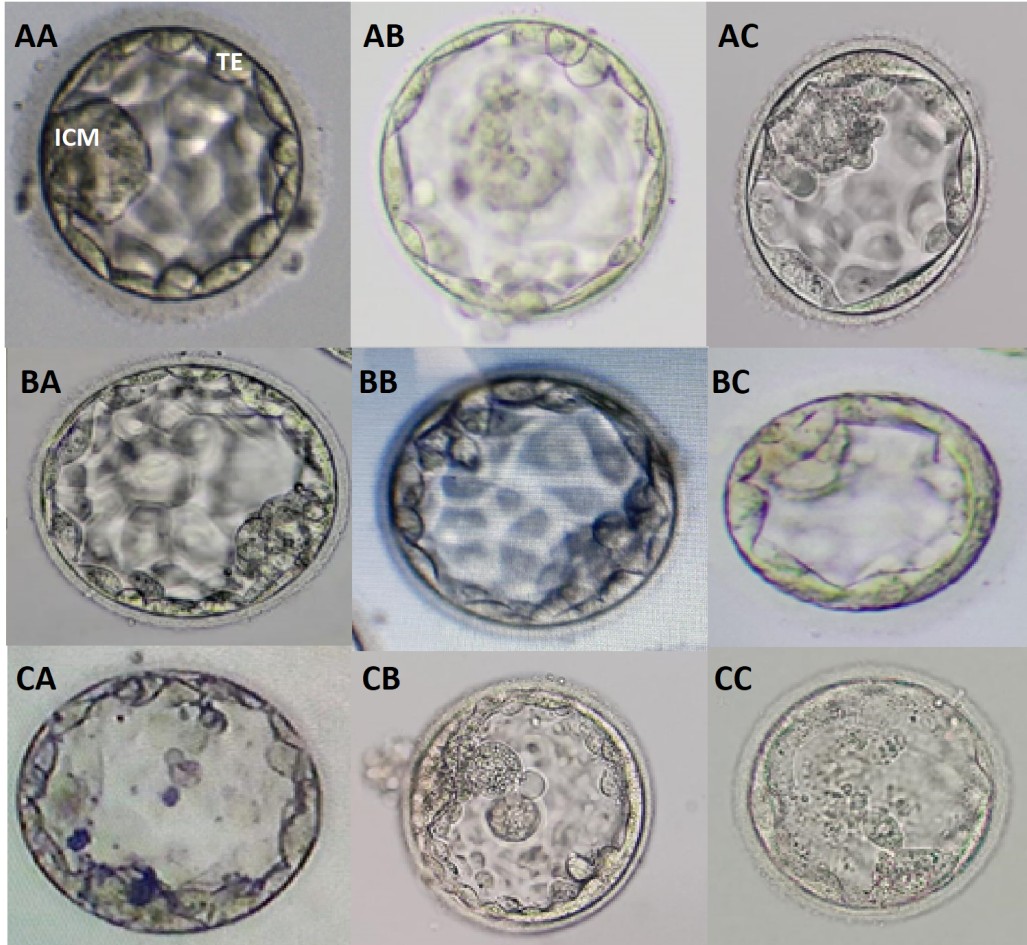

**Figure 1  Blastocyst grading based on quality of ICM and TE.** Different grades of blastocysts with possible combinations of ICM and TE morphologies are provided. AA, good graded ICM and TE; AB, good ICM with average TE; AC, good ICM with poor TE; BA, average ICM with good TE; BB, average ICM and TE; BC, average ICM with poor TE; CA, poor ICM with good TE; CB, poor ICM with average TE; CC, poor ICM and TE.

## Embryo thawing and embryo transfer

Blastocysts were thawed using vitrification warming medium (Kitazato, Japan) three hours before replacement to achieve a complete re-expansion of the blastocoel cavity. Our standard policy was the replacement of two blastocysts per transfer cycle but SETs were also facilitated where only a single blastocyst was available.

## Clinical outcome measures

The primary outcome measure was the clinical pregnancy characterized by the presence of gestational sacs with foetal heartbeat after 6–8 weeks after embryo transfer. Ongoing pregnancy and live birth rates were considered as secondary outcome measures.

## Statistical analysis

Statistical analysis was done using MEDCALC (Belgium) and SAS 9.4 software. Descriptive and inferential statistics were used to analyse the data. A comparison of pregnancies from different ICM/TE grades (AA, AB, BA, BB, CA, CB, and CC) was performed with the use of chi-square tests, Fischer's exact tests, Wilcoxon rank-sum test, and Mantel–Haenszel tests. Overall IR rates were expressed as percentages. Differences were considered significant when $p < 0.05$.

Logistic regression was applied to do the probability plot and analyse the relationship of the outcome to one or more predictors, the quality of ICM and TE. The area under the curve–receiver operating characteristic (AUC-ROC) was plotted between sensitivity versus specificity to measure the details of the interaction effect between ICM and TE, the two characteristics of the same embryo. In addition, the rank biserial correlation was used to find the correlation between dichotomous or nominal data and ordinal data. It is a special case of Somers' D.

## RESULTS

A total of 1,972 patients were enrolled in the study with a total of 3786 blastocysts transferred. Patients with single foetal cardiac activity from the transfer of two blastocysts of different grades who miscarried, and had no follow-up were excluded ($n = 819$) as explained in the patient inclusion and analysis flowchart (Fig. 2). The impact of morphology on miscarriage rate was not studied as most of them were DETs with blastocysts of heterogeneous grades.

Out of 1,153 KID patients included in the analysis, 352 were pregnant (CPR = 31%). In this group, CPR for women who had SET, DET of similar grades with a singleton pregnancy, and DET with twin pregnancy was 48% ($n = 43/90$), 49% ($n = 77/156$), and 26% ($n = 232/907$) respectively.

Out of 2,060 KID embryos, 584 (28%) resulted in implantation and 1,476 did not. IR from transfers of AA, AB, BA and BB were 48.5% ($n = 360/1102$), 39.4% ($n = 145/513$), 23.4% ($n = 22/116$) and 25% ($n = 55/275$) respectively. IR from BC was 4.5% ($n = 2/44$) whereas CB and CC grades showed no implantation with an inconsiderable size of transferred embryos. A statistically significantly higher IR was observed in blastocysts with ICM grade A ($p = 0.000027$ ($p < 0.001$)) than ICM B irrespective of TE morphology (Table 1).

Using a logistic regression model, the ROC curves were plotted. The AUC values for ICM, TE, and their interaction for pregnancy prediction were observed as 0.56, 0.55, and 0.57 (Fig. 3) respectively. In addition, the probability of good, average, and poor grades of ICM and TE and their association in determining the outcome was measured (Figs. 4 and 5).

The logistic regression analysis model fit showed a significant effect of ICM and TE quality in influencing the pregnancy outcome ($p < 0.0001$). The likelihood of pregnancy from each analyzed parameter was also estimated (Table 2). The estimated biserial correlation coefficient for ICM and TE using the ranking method was 0.11 and 0.05 respectively. This indicates a stronger association of ICM morphology with pregnancy than TE.

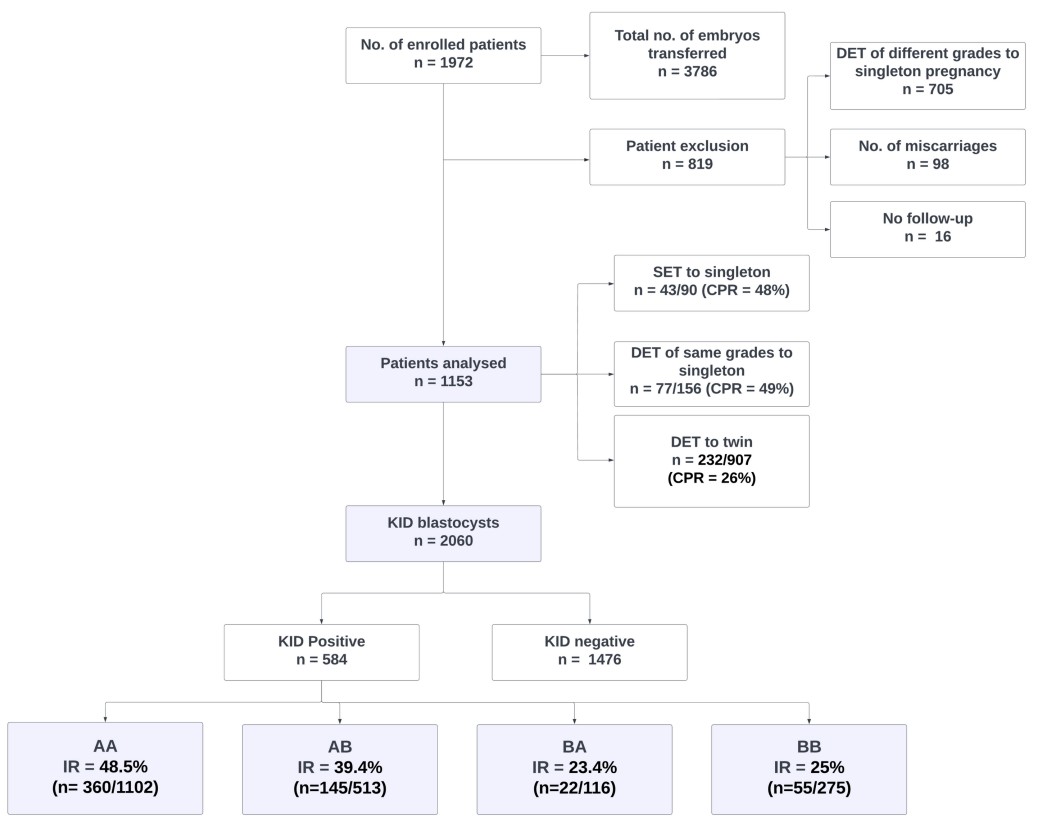

**Figure 2 Patient inclusion, exclusion and analysis flowchart.** DET, Double embryo transfer; SET, Single embryo transfer; KID, Known implantation data; AA, AB, BA and BB, Inner cell mass/trophectoderm grades of blastocysts.

The above results confirmed the superiority of ICM over TE in predicting the outcome of an IVF cycle. Though better TE grades were associated with a higher IR, ICM serves as a major indicator of blastocyst viability.

## DISCUSSION

This study demonstrates the benefit of conventional morphological assessment in the selection of the most appropriate blastocysts for replacement. We found that grades of both ICM and TE independently influenced the implantation. A significantly higher IR was observed with the transfer of better grades of ICM than those of TE. The model prediction was suggestive of ICM as a stronger predictor of a successful outcome.

As stated in previous reports, the present study also indicated a direct correlation between the morphology and the implantation. Blastocysts with top grades of ICM showed an increased IR (AA = 48.5% and AB = 39.4%) comparing the average grades (BA = 23.4% and BB = 25%) irrespective of TE quality. In other words, TE, only in the presence good quality ICM, showed a better IR. This confirms that the ICM quality determines the viability of a blastocyst than TE.
**Table 1 Implantation rates according to the grades of ICM and TE.** The pregnancy outcomes of the patients according to the morphological grades of ICM and TE were compared for significance.

| S.no. | ICM Grade | TE Grade | Pregnancy | | Pregnancy percentage(%) |
|---|---|---|---|---|---|
| | | | Positive | Negative | |
| 1 | A | A | 360 | 742 | **48.5** |
| 2 | A | B | 145 | 368 | **39.4** |
| 3 | A | C | 0 | 4 | 0 |
| 4 | B | A | 22 | 94 | **23.4** |
| 5 | B | B | 55 | 220 | **25** |
| 6 | B | C | 2 | 44 | 4.5 |
| 7 | C | A | 0 | 0 | 0 |
| 8 | C | B | 0 | 2 | 0 |
| 9 | C | C | 0 | 2 | 0 |
| | | $\chi 2$ test (S.no. 1,2,4,5) $= 23.820$; $df = 3$; $p < 0.001$ ($p = 0.000027$)* | | | |

**Notes.**

**Tests used**: chi-square tests, Fischer's exact tests, Wilcoxon rank-sum test, and Mantel–Haenszel tests.

ICM, Inner cell mass; TE, trophoectoderm; A, Good; B, Average; C, Poor.

*Statistically significant

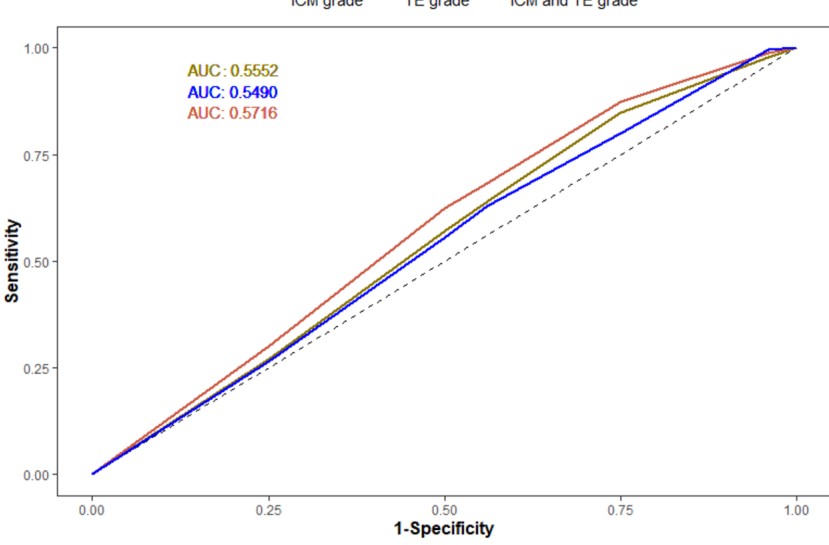

**Figure 3 Logistic regression model for ICM, TE and their interaction and pregnancy prediction.** (1) The green line illustrates the AUC–ROC curve for ICM. (2) The blue line illustrates the AUC–ROC curve for TE. (3) The red line illustrates the AUC–ROC curve for the interaction between ICM and TE in predicting the pregnancy.

From the logistic regression analysis, the measured AUC values for ICM and TE suggested no discrimination between their predictive ability of pregnancy. This can substantiate the statement mentioned above that both could have an impact on implantation. The further analysis of their interrelation using ROC curve and biserial correlation interpreted TE's dependency on ICM for a successful implantation.
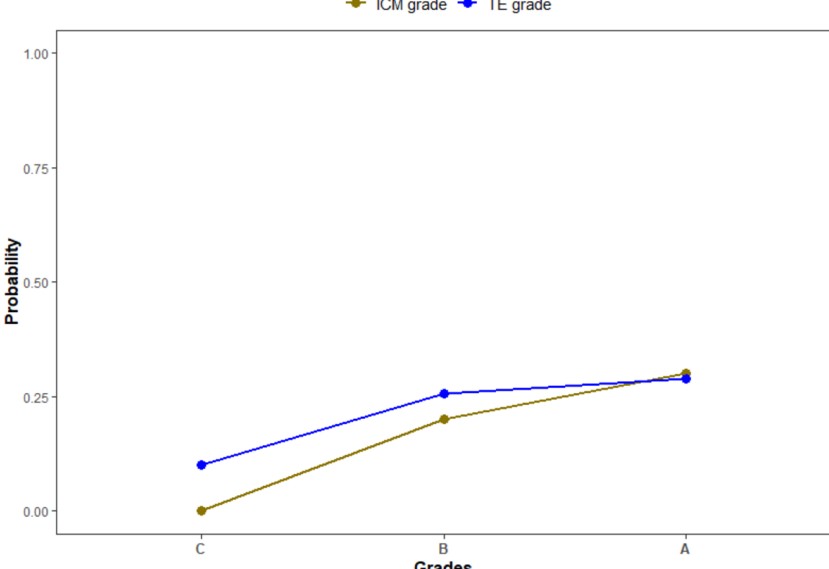

**Figure 4 The pregnancy prediction of the morphological grades of ICM and TE using logistic regression model.** (1) The green line illustrates the predictive probability of different grades of ICM (A—good, B—average and C—poor) (2) The blue line illustrates the predictive probability of different grades of TE (A—good, B—average and C—poor).

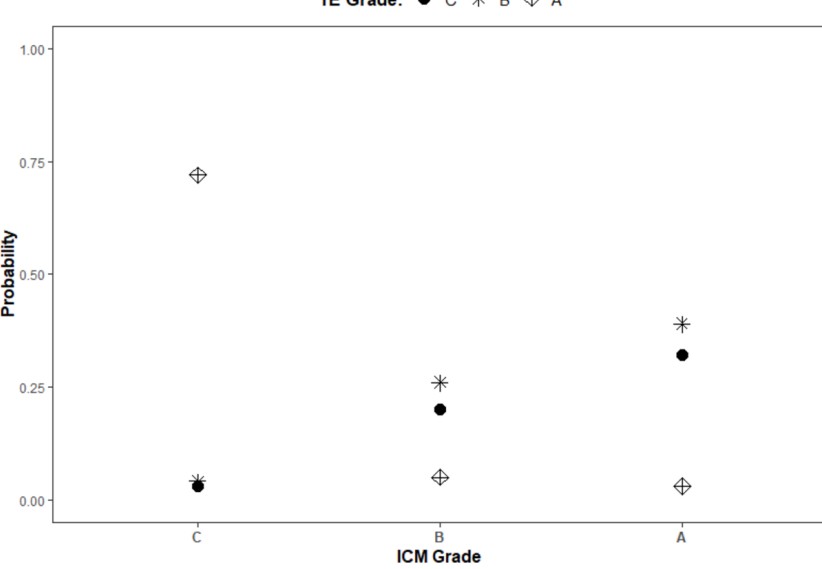

**Figure 5 Logistic regression model for the interaction of ICM and TE morphology in pregnancy prediction.** (1) ICM morphology, A—good, B—average, and C—poor, as plotted on the $X$-axis. (2) TE grades, A—good, B—average, and C—poor, illustrated by different shapes in the graph and explained in the title.

**Table 2 Estimation of predictive ability of morphological grades of ICM, TE and their association using logistic regression model.** (A) ICM: Inner cell mass. (B) TE: Trophectoderm (C) B: Average grade. (D) C: Poor grade.

**Analysis of maximum likelihood estimates**

| Parameter | | | DF | Estimate | Standard Error | Wald Chi-Square | Pr >ChiSq |
|---|---|---|---|---|---|---|---|
| Intercept | | | 1 | −4.9247 | 85.1290 | 0.0033 | 0.9539 |
| ICM | C | | 1 | −7.5472 | 170.3 | 0.0020 | 0.9646 |
| ICM | B | | 1 | 3.4136 | 85.1290 | 0.0016 | 0.9680 |
| Intercept | | | 1 | −1.6942 | 0.2426 | 48.7549 | 0.0(<.0001) |
| TE | C | | 1 | −1.5233 | 0.4816 | 10.0029 | 0.0016 |
| TE | B | | 1 | 0.6123 | 0.2472 | 6.1369 | 0.0132 |
| Intercept | | | 1 | −1.8095 | 0.1973 | 84.1241 | 0.0(<.0001) |
| ICM*TE | C | C | 1 | 2.7153 | 0.7132 | 14.4960 | 0.0001 |
| ICM*TE | C | B | 1 | −1.2873 | 0.3956 | 10.5907 | 0.0011 |
| ICM*TE | B | C | 1 | −0.7684 | 0.3658 | 4.4116 | 0.0357 |
| ICM*TE | B | B | 1 | 0.4211 | 0.2166 | 3.7787 | 0.0519 |

**Association of Predicted Probabilities and Observed Responses**

| | | | |
|---|---|---|---|
| Percent Concordant | 37.7 | Somers' D | 0.143 |
| Percent Discordant | 23.4 | Gamma | 0.234 |
| Percent Tied | 38.8 | Tau-a | 0.058 |
| Pairs | 861984 | c | 0.572 |

Both ICM and TE play a significant role in establishing a viable pregnancy. The functions of TE involves forming the placenta whereas the ICM is intended to form the foetus as well as the yolk sac, the extraembryonic mesoderm, the amnion, and the allantois. This could attribute to ICM being an effective predictor of the clinical outcome.

As proposed in this study, grades of ICM are positively correlated to a viable pregnancy and ICM grade A should be prioritized for transfer. Our findings were consistent with various other studies. A recent report from *Ai et al. (2021)* evaluated the association of the morphological parameters in frozen SETs and found ICM grade was a stronger predictor of live birth (*Ai et al., 2021*). Their classification of blastocysts and the conclusions were similar to the current study and may support our findings. *Subira et al. (2016)* found that better ICM quality could help in improving the likelihood of live birth rates in fresh SET cycles. The study by Evans in 2021 showed increased CPR from the transfer of blastocysts with higher grades of ICM than TE (*Evans & Knaggs, 2021*). This could be supported by the studies analysing the other parameters in addition to visual estimation of blastocysts such as 'the ICM to blastocyst diameter ratio' (*Almagor et al., 2016*).

The ICM grades in the prediction of euploid and aneuploid miscarriages have also been investigated. A study by *Van den Abbeel et al. (2013)* showed that using blastocysts with grade A ICM resulted in lower rates of pregnancy loss. Another study evaluated the potential of using ICM scores to predict rates of miscarriage (*Shi et al., 2020*). *Licciardi et*

*al. (2015)* explained that birth weight was associated with the morphological feature, the ICM (*Licciardi et al., 2015*).

The implications of the findings in this study in terms of PGT are still to be evaluated. Irani et al. studied the positive association between ICM grades and ploidy status from the transfer of euploid single blastocysts. This clearly indicates that ICM morphology can reflect the ploidy of blastocyst (*Irani et al., 2017*; *Nazem et al., 2019*).

Various other studies have investigated the positive correlation between TE morphology and an increase in the pregnancy rates (*Chen et al., 2014*; *Ahlström et al., 2011*; *Honnma et al., 2012*) but the importance of the quality of ICM was not underestimated. *Yoshida et al. (2018)* studied the influence of TE morphology on ploidy status but suggested that the quality of TE was not the single indicator of ploidy of blastocysts.

Even though some studies discussed TE as a stronger predictor, ICM prediction seems to have garnered more significance. This is supported by the use of ROC curve analysis in the current study.

The strength of this study was the use of a simple, clinically validated, non-invasive, easily reproducible, and cost-effective method of evaluation of blastocysts to predict successful pregnancies in IVF. This selection strategy can be interpreted to encourage SETs.

The main limitation of this study was no other variables other than standard morphology were compared with the clinical outcome. It would have been more useful if the impact of the key morphological features was studied with SETs. This study can be expanded to comparing live birth outcomes of the same cohort of patients to reassess the current findings.

## CONCLUSIONS

There is a current deliberation in terms of the morphological characters involved in embryo assessment and selection as various studies have independently advocated the use of ICM and TE morphology as valuable predictors of pregnancy and live birth. The findings of our study suggest ICM is the strongest predictor of blastocysts with the highest probability of implantation continuing to a clinical pregnancy compared to TE and is further proved with statistical analysis. The current selection method based on morphology can still be used as a gold standard for improved IVF success.

The other advanced invasive and non-invasive approaches for the identification of novel viability markers are still under investigation. These independent markers, in adjunct with morphology, can be employed to represent the embryo quality, once demonstrated with their potential in infertility management.

## ACKNOWLEDGEMENTS

The authors would like to thank the embryology team and our statisticians of ARC International Fertility and Research Centre for their expert technical assistance.

### Funding
The authors received no funding for this work.

### Competing Interests
The authors declare there are no competing interests.

### Author Contributions

- Sargunadevi Sivanantham conceived and designed the experiments, performed the experiments, analyzed the data, prepared figures and/or tables, authored or reviewed drafts of the article, and approved the final draft.
- Mahalakshmi Saravanan performed the experiments, authored or reviewed drafts of the article, and approved the final draft.
- Nidhi Sharma conceived and designed the experiments, authored or reviewed drafts of the article, and approved the final draft.
- Jayashree Shrinivasan performed the experiments, authored or reviewed drafts of the article, and approved the final draft.
- Ramesh Raja conceived and designed the experiments, performed the experiments, analyzed the data, prepared figures and/or tables, and approved the final draft.

### Human Ethics
The following information was supplied relating to ethical approvals (i.e., approving body and any reference numbers):

Institutional Ethics Committee of Saveetha Medical College Hospital (SMCH), Chennai India granted Ethical approval to carry out the study in the ARC Interantional Fertility and Research Centre facility, Chennai, India (Reference Number- 006/09/2020/IEC/SMCH).

### Data Availability
The raw data is available in the Supplemental Files.

### Supplemental Information
Supplemental information for this article can be found online at http://dx.doi.org/10.7717/peerj.13935#supplemental-information.

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
