# Peer review of "Morphology of inner cell mass: a better predictive biomarker of blastocyst viability"

_PeerJ, doi:10.7717/peerj.13935_

## Round 0.1 · original submission · Minor Revisions

Your manuscript could be accepted after incorporating the suggestions by both reviewers.

Reviewer 1 ·

Basic reporting

Minor comments/editorial
L54 comprises instead of composes of
L55 In routine practice,…
L63-64 this sentence makes no sense
L80-81 unexplained abbreviations
L118 metabolite not metabolome
L118-121 is not a proper sentence: add “is” before relevant?
L123-4 …need to be considered before omics application….
L222-224 unexplained (in text) abbreviations
L242 this statement is unclear.
Figures 2 onwards and tables. These should be condensed into one or two tables or figures with legends as largely statistical analyses.

Experimental design

This study represents a large amount of data accumulated in various clinics over the course of a year and claims that a high ICM grade is more predictive of successful 8 week pregnancy than a high TE grade with grading based on a simple morphological scheme. While the data itself appears convincing there are concerns about reproducibility, in that 1. the grading process is poorly described and seems badly controlled, and 2. it seem the numerical 1-6 classification was ignored for analytical purposes. This is disconcerting as the developmental age (1-6) will affect the number of TE and ICM cells disproportionally and thus may bias the results. It is also unclear how different embryologists compared their grading to be consistent and whether the two involved in grading any embryo were doing so blindly of each other. The data could be analysed by including “embryologist” as a term in the statistical analysis.
L180 Where is FigS4? I also checked whether from ref Gardner 2004 but this is not the case either. Please supply these pictures.
L187 Methods: Were photos taken of graded blasts for embryologists to compare or did one embryologist grade and sort embryos for cryopreservation with second embryologist checking? What is meant by “the highest available grades of them were selected”? Does this refer to the 1-6 grading system or the ABC/ABC ICM/TE grading?
Figures 2 onwards and tables. These should be condensed into one or two tables or figures with legends as largely statistical analyses.

Validity of the findings

L252-8 Diczfalusy’s concept is not likely relevant as too late in pregnancy (outcome measured was 6-8 weeks after embryo transfer). The ICM gives rise not only to the embryo proper (foetus) but also to the yolk sac, the extraembryonic mesoderm, the amnion and allantois. The extraembryonic mesoderm in particular gives rise to the foetal vasculature of the placenta from week 2.

Reviewer 2 ·

Basic reporting

Must improve the English language for clarity. Many sentences are too long and can be split to improve the overall manuscript's readability. Example; lines 267- 272.

Should cite all references as per the PeerJ guidelines. Exclude redundant references 47 and 48.

Provide high-resolution figures with a detailed legend for each figure. The legend should start with a title followed by a description, which is missing in the current script.

Figure 2 (flow-chart) is not clear in the PDF file provided.

Figures 5-7 annotation errors.

Figures 3-8 can be merged into one or two figures.

Tables 2, 3, 4, 6, 8, 11, and 12 are not needed, as the values in these short tables could be moved to the results and discussion part wherever appropriate.

Enclose all the remaining high-quality images of the blastocysts, if available, in the supplementary information along with other data as it may be helpful for the readers.

The discussion part needs more elaboration. For example, lines 248-250.

Experimental design

No comment.

Validity of the findings

The statistical data on miscarriage rate and ploidy status may be included, if available, to further validate the finding of this manuscript.

Additional comments

The methodology and the conclusions drawn in this manuscript (pages 174-187 and 293-299, respectively) are very similar to the already published article (https://doi.org/10.3389/fendo.2021.621221). A comparative statement on the findings from this manuscript and the one cited here would be really useful.

As this study focuses on the south Indian population, the authors may want to incorporate the phrase "...in south Indian population" in the title to reflect this.

---

## Round 0.2 · accepted · Accept

The authors have paid careful attention to the reviewers' comments. The manuscript has been substantially revised in accordance with the comments of the reviewers. If we need to clarify any details required to move the manuscript forward, then our production staff will get in touch with you.